# Benefits of a Three-Day Bamboo Forest Therapy Session on the Physiological Responses of University Students

**DOI:** 10.3390/ijerph17093238

**Published:** 2020-05-06

**Authors:** Chengcheng Zeng, Bingyang Lyu, Songyuan Deng, Yan Yu, Nian Li, Wei Lin, Di Li, Qibing Chen

**Affiliations:** College of Landscape Architecture, Sichuan Agricultural University, Chengdu 611130, China; zcclandscape@163.com (C.Z.); beyonglv@163.com (B.L.); dsy20101802@163.com (S.D.); 15682010818@163.com (Y.Y.); nli@sicau.edu.cn (N.L.); landscape1990@163.com (W.L.); frank0707@126.com (D.L.)

**Keywords:** bamboo forest therapy, environmental evaluation, physiological response

## Abstract

Studies have indicated that natural environments have the potential to improve the relationship between a stressful life and psychological well-being and physical health. Forest therapy has recently received widespread attention as a novel solution for stress recovery and health promotion. Bamboo is an important forest type in many countries, especially in East and Southeast Asia and in African countries. Bamboo is widespread throughout southwestern China. Empirical field research on the physiological effects of bamboo forest therapy is currently lacking. To explore the benefits of bamboo forest therapy on the physiological responses of university students, 120 university volunteers between the ages of 19 and 24 participated in this study (60 males and 60 females) and were randomly divided into four groups of equal size (15 males and 15 females in each). Four sites were selected for the experiment, including two natural bamboo forests (YA and YB), a bamboo forest park (DJY), and an urban environment (CS). During the testing period, all participants were asked to view the landscape for 15 min in the morning and then walk in the testing area for 15 min in the afternoon. Blood pressure (BP), heart rate (HR), and peripheral oxygen saturation (SpO2) were measured as the physiological indexes, and the semantic differential method (SDM) questionnaire was completed for the environmental satisfaction evaluation. The SDM for the subjective environmental evaluation differed significantly among the university students; they obtained a better environmental experience, in terms of sensory perception, atmosphere, climate, place, and space, in the bamboo forest sites. The three-day bamboo forest therapy session improved the physiological well-being of university students. First, the blood pressure and heart rate of the university students decreased, and the SpO2 increased, after the three-day viewing and walking activities of the three-day bamboo forest therapy session. The viewing activities had a more pronounced effect on decreased heart rate in university students. Additionally, three-day bamboo forest therapy had a positive impact on decreased systolic blood pressure and heart rate in the university students, and it was significantly decreased in females, while peripheral oxygen saturation (SpO2) remained relatively low. Finally, compared with the urban site (CS), the bamboo forest sites effectively improved the university students’ physiological state of health, decreased their physical pressure, and stabilized their physiological indicators. These findings provide scientific evidence that a three-day bamboo forest therapy session can increase positive physiological responses. The potential for a longer-term effect on human physiological health requires further investigation.

## 1. Introduction

Global urbanization comes with health threats to populations [1]. The most important factor is the deterioration of the urban living environment. Evidence has shown that people suffer psychological and physiological pressure in the urban environment. Air pollution, noise pollution, water pollution, work pressure, and other stresses related to urban environments are increasingly compelling humans to seek forms of stress relief and healthy lifestyles [2]. There are many positive aspects of urban life, such as employment, higher incomes, better opportunities for education, and access to health care, encourage rural to urban migration [3]. These attracted people to prefer living in large cities rather than rural areas [4]. Many studies have shown that excess artificial environmental stimulation may have negative effects on human health [5]. Although urban green spaces offer residents important opportunities for recreation and restoration, in the countries with large populations they are difﬁcult to defend when faced with various land-use interests [6]. Research has shown that high blood pressure (BP) costs the U.S. approximately $48.6 billion per year and affects 1 in 3 Americans [7]. University students have several health risk factors, including irregular sleep patterns, personal relationship changes, excessive drinking, and academic pressures, and they experience a large amount of stress, anxiety, and depression [8]. According to repeated previous studies, approximately 50% of university students experience signiﬁcant levels of stress, anxiety, depression, or a combination of these conditions [9]. Various physiological and psychological diseases are caused by stress, thus affecting human well-being and health [10].

Because of the negative environmental impacts in urban areas, research on the benefits of immersion in the natural environment is important. In most humans, the relationship between survival and the natural environment is inseparable. Current evidence has shown that nature has great benefits for the human brain, including increased happiness, health/well-being, and cognition [11]. In recent years, increasing attention has focused on the role of forests in promoting human health and well-being [12]. Some theories about environmental health have been proposed, including the attention restoration theory [13,14] and Ulrich’s “Stress Recovery Theory” [15] which predict that natural scenes tend to reduce stress, whereas settings in the built environment tend to hinder recovery from stress. Similar studies confirmed that natural environments have the potential to improve the relationship between a stressful life and psychological well-being and physical health [16,17,18,19].

Forest therapy has been proposed as one of the solutions for stress recovery and health promotion [20] and has recently received widespread attention as a novel approach to achieving physiological and psychological relaxation. Beginning in 2004, Li Q and colleagues [21] conducted a series of studies on the effect of forest therapy on human health and established forest therapy as a new preventive strategy. Forest therapy has many beneficial effects on human physical, psychological, and immune systems.

Other researchers have sought to improve the description and evaluation of the relationship between forests and human health [22]. Yu [10] attempted to conduct field studies on forest therapy and provided interesting scientific data supporting the hypothesis that physiological indexes, such as BP, pulse rate, and heart rate variability, are impacted by forest therapy. Tsunentsugu Y and Park BJ [10,23] studied various physiological indexes, such as cerebral activity in the prefrontal area, heart rate variability (HRV), pulse rate [23], and blood pressure [24,25], and they provided evidence regarding the relatively positive effects of forest environments on human physiological activities compared with the effects of artificial environments. Moreover, a forest therapy program led to significantly higher parasympathetic nervous system activity [26] and lower sympathetic nerve activity [23] than urban areas. Additionally, some research has shown that, compared with the urban environment, forest therapy was capable of increasing positive mood states and reducing negative mood states and specific psychological responses [10,25,26,27]. Park BJ [28] conducted a semantic differential method (SDM) in which a questionnaire was administered to subjects prior to their walks in forests and urban areas. Responses indicated that forest settings are perceived as being significantly more enjoyable, friendly, natural, and sacred than urban settings. Overall, previous research has provided a large amount of evidences that forest environments have beneficial effects on health.

However, these recent studies have paid less attention to the effects of different tree forests on human health. Bamboo forests are a major type of non-wood forest [29] and an integral part of forestry; they are widely distributed in East and Southeast Asia and in African countries. Bamboo forests are a versatile and important component of the ecology, culture, and economy of these countries [30]. Bamboo forests in different countries and contexts have been proven to be the best option for both landscape restoration and the supply of various ecosystem services. These forests supply more ecosystem services than any other type of planted forest [31]. Especially in recent decades, the forest area of the world has been gradually reduced, while bamboo forests have increased in area by 3% every year. Bamboo is well-known and the most preferred plant in Chinese landscape design because of its unique beautiful foliage and fast-growing characteristics. A study of bamboo forest therapy [32] showed that the bamboo forest environment is more natural, comfortable, open, and bright with pleasing ambient noise and that it was better for psychological and physical relaxation among male college students than urban sites. Ye Zhang [33] studied the negative oxygen ion concentration of 18 species of ornamental bamboo and showed that in three seasons, diurnal variation in negative oxygen ions was high in the morning, decreased at noon, and gradually increased in the evening. The concentration of negative ions in most bamboo species was significantly higher in summer and autumn than in spring. Ahmad Hassan [34] found that viewing bamboo plants resulted in significantly lower systolic and blood pressures but no changes in the pulse rate among 40 university students. In another study by Ahmad Hassan [35], a 15-minute walk in a bamboo forest was found to improve mood and reduce anxiety.

However, to our knowledge, empirical field research on the physiological effects of bamboo forest therapy is currently lacking. We hypothesized that bamboo forest therapy would provide benefits on the psychophysiology of humans similar to the effects of forest therapy. Therefore, the aim of the present study is to investigate the physiological health benefits of three-day bamboo forest therapy among university students. A total of 120 university volunteers participated in this study (60 males and 60 females) and were randomly divided into four groups of equal size (15 males and 15 females in each). All participants were asked to view the landscape for 15 min in the morning and then walk in the testing area for 15 min in the afternoon. Blood pressure, heart rate, and peripheral oxygen saturation (SpO2) were measured as the physiological indexes, and the semantic differential method (SDM) questionnaire was completed for the environmental satisfaction evaluation. This study will first discuss the results of the environmental evaluations in the sequence of the semantic differential method (SDM), followed by a discussion of the results of the relationship between environmental factors and environmental evaluations. Finally, the physiological responses will be compared between males and females when viewing and walking to provide scientific evidence for the physiological health benefits of three-day bamboo therapy.

## 2. Materials and Methods

### 2.1. Subjects and Experimental Sites

All experimental sites used in this study were located in Sichuan Province in Southwest China. The city site (CS) in Chengdu was located in the center of downtown (30°42’33.2”N 103°51’59.4”E). In Chengdu, the subjects could view urban buildings, cars, people, and other urban elements. For the first bamboo forest site, a site located near the city of Ya’an (YA) was selected (29°54’18.9”N 103°02’05.3”E). The dominant forest species of the selected site (and Sichuan Province) is *Neosinocalamus affinis*, a large species of cluster bamboo. The Yibin (YB) site (28°28’22”N, 105°0’19”E) was located in Yibin City, which is well-known for the Shunan Bamboo Sea forest, an AAAA-level scenic area, as denoted by the Chinese National Tourism Administration. The Shunan Bamboo Sea comprises more than 12,000 ha of large *Phyllostachys heterocycla*. The Dujiangyan (DJY) site (31°44’54”N, 103°25’42”E ) is a bamboo park in Dujiangyan City named ‘Zhuhai Dongtian’, which is covered with *P. praecox f. prevernalis* bamboo (C. D. Chu et C. S. Chao XXX). Figure 1 shows the locations of the four research study sites in Sichuan Province for our study, and Figure 2 shows photographs of the four sites [32].

In this study, 120 university students participated in three-day field experiments. None of the participants reported any physiological or psychiatric disorders in their personal histories. Subjects who smoked or had a history of abusing alcohol were excluded from this study. Before the experiment, the goal and experimental procedures of the study were explained to the participants, and informed consent was obtained. This study was reviewed and approved by the Ethics Committee of Sichuan Agricultural University. To control the background environmental conditions, identical single rooms, and similar meals were provided to each subject for the duration of the study period. The subjects were randomly divided into four groups, and each group included 30 university students. Besides, the basic characteristics of the participants, such as the age, height, and weight of each subject, were measured. To avoid physiological differences, the systolic blood pressure (SBP), diastolic blood pressure (DBP), heart rate (HR), and blood oxygen saturation levels of each participant were measured. After collecting the basic data of the participants, the analysis of variance was used for the data comparison between the CS and bamboo forest sites. The results showed no significant differences between the CS and bamboo forest sites in the subjects’ age, height, or weight. The results are shown in Table 1 [32].

Noise, air temperature, absolute illumination, relative humidity, radiant heat, negative air ionization, and wind velocity were measured at each experimental site. A commercially available PMV meter (AM-101) was used to measure the air temperature, relative humidity, radiant heat, and wind velocity. A digital photometer (T-1H) was used to measure the lighting. An acoustic environment factor was used to measure the noise (CEN-TER322). Negative oxygen ions were measured with AIC 2000. All the measurements were recorded every 1 h at each experimental site from 9:00 am to 5:00 pm, and then the average was used for assessment. Compared with the CS site, the bamboo forest sites showed significant differences in temperature, relative humidity, radiant heat, noise, absolute illumination, and wind velocity (Table 2).

### 2.2. Procedures

We chose to conduct this study in September, as it is a suitable month for outdoor travel. The weather was sunny during the experimental period. On the afternoon of September 19th, the subjects were divided into four groups. The groups arrived at the arranged hotels near the given experimental sites. The experimental sites were flat areas where the experiment could be easily conducted. All hotels were located in the bamboo forest and urban approximately 300 m from the experimental site. To prevent the influence of subjective psychological effects, the subjects were allowed to do as they wished in the hotel but were instructed to avoid strenuous exercise and any stimulating activity in their hours of relaxation before sleeping. On the morning of September 20th, 2017, the first day of the experimental period, each group was taken to its respective experimental site after a normal breakfast. The subjects sat on chairs and viewed the landscape for 15 min, then the physiological data were sampled. They walked around the area for 15 min in the afternoon, physiological data and SDM questionnaires were sampled. The course terrain, distance, and walking speed were appropriate for moderate energy expenditure. After completing the walking and viewing activities, the participants were allowed to move freely within the experimental sites until mealtime. All of the subjects were instructed to remain at their assigned experimental site from 9:00 am to 5:00 pm, except during lunchtime. After the first day, the same procedure was conducted for the next two days (Figure 3).

### 2.3. Measurements

(1) Semantic differential method (SDM).

Psychological reactions were investigated based on the subjective evaluation SDM (semantic differential method) [36], which is an environmental space evaluation method that has been verified to be scientific and practical. Besides, this method allows for the quantification of the respondents’ evaluation of their experience of the environmental space, and the leading factors influencing the environment can be obtained by combining reasonable mathematical analysis methods to draw more scientific and reasonable conclusions regarding the environmental factors influencing human psychology. Through the questionnaire, which includes 20–30 pairs of negative adjectives, the test subjects’ subjective impressions of the environment were rated according to five grades: “−2”, “−1”, “0”, “1” and “2.” The measurements were recorded for one subject at a time after the “Walking” phase of the study between 03:30 pm and 03:45 pm. In this study, a total of 360 SD questionnaires were issued. After they were completed, invalid and incorrect questionnaires were removed. Finally, 348 valid questionnaires were recovered, with an effective rate of 96.67%.

(2) Physiological Indexes.

SBP and DBP were measured with automated BP devices (Omron HEM-7112 Comfort). BP was measured on the left arm, the measurements were conducted after 10 min rest in a seated position, and three measurements were taken at 30 s intervals. Peripheral oxygen saturation (SpO2) was measured three times with a pulse ox meter (Philips DB18, Suzhou, China) on the index finger of the left hand. HR was measured three times with a single-channel electrocardiograph (Med-ECG-2301, Guangzhou, China) [32].

### 2.4. Statistical Analysis

The bamboo forest data were from the Ya’an, Yibin, and Dujiangyan sites. For both the urban and bamboo forest sites, the data from the SDM questionnaire after walking were averaged across 3 days in the afternoon. The data from the SDM questionnaire for the four experimental sites were averaged across 3 days. For the comparison of SDM scales between bamboo forest and urban setting, KMO and Bartlett test were used. The principal component analysis was performed to analyze the sensory evaluations reported using the SDM. The differences between viewing and walking activities between the two sexes (i.e., male and female) at the same experimental site were evaluated using an independent paired sample t-test responses to a walk with an alpha level of 0.05. Otherwise, one-way ANOVA (Turkey’s test) was used to test for the differences among experimental sites for the same gender in each variable. The Statistical Package for Social Sciences software (v20.0, SPSS Inc., Chicago, IL, USA) was used for all statistical analyses. SDM data are presented as the means ± standard deviation (SD), physiological data are presented as the means ± standard errors (SEs), and differences were considered significant at *p < 0.05*. Origin9 (Origin Lab Inc., Northampton, MA, USA) was used to plot the analysis results.

## 3. Results

### 3.1. There Was a Significant Difference in Subjective Environmental Assessment (SDM) between the Bamboo Forest Environments and Urban Environment

The comprehensive environmental characteristics of the bamboo forest (BF) sites were compared with those of the urban environment (CS) in an environmental evaluation (Figure 4).

Results from the evaluations of the psychological responses to forest and urban environments reported using the SDM after the “Walking” phase of the study. Data are presented as the means ± SD; *n* = 348; *p <* 0.05.

Figure 4 shows data from four walking events in the bamboo forest and urban environments for 120 students. Compared with the urban environment, the average SDM bar chart of the test results shows that the bamboo forest environment was reported to have higher scores on the following indicators: “natural” (in terms of relative and artificial), “health” (as opposed to unhealthy), “beautiful” (relative to ugly), peace of mind” (relative to disturbing), “happy” (relative to unhappiness), “dark” (relative to too bright), “quiet” (relative to noisy), “interesting” (relative to boring), “harmonious” (relative to unrest), “comfortable” (relative to uncomfortable), “multi-level” (relative to the surface), “friendly (relative to unfriendly), “charming” (relative to less glamorous), “cold” (relative to the warm), “security” (relative to dangerous), “wet” (relative to dry), “aromatic” (relative to stench), “characteristic” (relative to no characteristics), “thick” (relative to thin), “orderly” (relative to chaos), and “no peculiar smell” (relative to a distinct smell). Compared with urban environments, the following descriptors were less often used in reference to the bamboo environment: “open” (as opposed to closed), “brighter” (as opposed to darker), and more “flowing” (as opposed to still). However, it was also found that the subjects’ SDM evaluation of the environmental characteristics corresponding to humidity did not meet the relative humidity value measured in the test, indicating that there might be differences between the subjective evaluations and the actual environmental characteristics. The subjects’ psychological evaluation of the environment was determined by multiple factors of the environment. Overall, compared to the urban site, the bamboo forest environment was significantly better in the subjects’ comprehensive evaluation.

#### Principal Component Analysis

The results of the principal component analysis are shown in Table 3. When the proper value was assumed to be 1 or more, the above table is generated by factor rotation and five factors were extracted. Five main factors can explain the bamboo forest environment characteristics of 25 analysis factor 61.28% internal relations. For the first principal component (F1), the factor load of 10 evaluation items is greater than 0.54. The highest rating of “beauty (Beautiful to ugly)” was 0.84, 10 evaluation index is shown by the investigation of bamboo forest environment characteristics of sensory factors, therefore, we defined the factor 1 (F1) as sensory factors. The factor load of 6 evaluation items was greater than 0.40, and the highest evaluation index of “aroma sensation (aromatic -malodorous)” was 0.72. As shown in the above six evaluation indexes obtained from the investigation of bamboo forest environment characteristics, the factor 2 (F2) was defined as the atmosphere factor. There are three evaluation of load factor where the value is greater than 0.53, for “cold (cold to hot)” the evaluation index was up to 0.71. It is shown in the above three evaluation index surveys of bamboo forest environment characteristics of climate factors, therefore, it was defined the factor 3 (F3) climate factor. The three-evaluation factor of the load is greater than 0.41; “flowing (flowing to stagnation)” evaluation index up to 0.77. The above three evaluation indexes show the environmental characteristics of the bamboo forest obtained from the survey, in which factor 4 (F4) is defined as the place factor. Three-evaluation factor of the load is greater than 0.53; “thickness (thick to thin)” the evaluation index of up to 0.69. Factor 5 (F5) was defined as a space factor.

Through factor analysis, five important influencing the factors are obtained as the main evaluation criteria of bamboo forest environment.

The comprehensive score of the five environmental characteristic factors is obtained (Table 4). YB (F = 1.01) > YA (F = 0.96) > DJY (F = 0.92) > CS (F = 0.09). The results show that the subjects for the comprehensive evaluation of the bamboo forest environment are significantly better than that of the urban environment, including the environmental evaluation of YB, highest subjects for YB overall environment of the highest recognition. 

After “walking,” according to the SDM sensual evaluation F scores, the sensory composite scores (F) to the environment and the environment temperature, relative humidity, thermal radiation, and wind speed, noise index, the absolute intensity of illumination, and negative oxygen ion correlation analysis results are shown in Figure 5. It is shown that the subjects of sensory evaluation and environmental temperature, radiant heat, wind speed, noise, the absolute intensity of illumination were significant, or significantly negatively correlated; the results show that the lower the environment variable, the higher the subjects for the environment of the sensory evaluation. The results showed that the higher the concentration of air negative oxygen ion, the higher the sensory evaluation of the environment.

### 3.2. Viewing Activities Had a Better Physiological Response than Walking

In all three kinds of bamboo forest environments, viewing and walking activities had positive effects on the physiological indicators of the subjects. The physiological indexes of the university students changed within the normal range. The data analysis shows that, compared with the urban environment (CS), there was a decrease in blood pressure after the viewing and walking activities in the bamboo forest environments. Especially in the Yibin bamboo sea scenic spot (YB) environment, SBP and DBP decreased by an average of (7.52 ± 1.13 mmHg; Figure 6a) and (3.22 ± 1.21 mmHg; Figure 6b) respectively. Systolic blood pressure showed a downward trend after the walking activity, and the mean systolic blood pressure in the three bamboo forest sites decreased at YA site (3.28 ± 1.21 mmHg; Figure 6a), YB site (4.45 ± 0.84 mmHg; Figure 6a), and DJY site (5.42 ± 0.74 mmHg; Figure 6a). Compared with the urban (CS) environment, the heart rate decreased after the viewing and walking activities in the bamboo forest environment. In the bamboo forest environment, the Yibin bamboo sea scenic site (YB) showed a decrease in heart rate. The average value decreased by (6.55 ± 1.24 bpm; Figure 6c) and (6.18 ± 1.17 bpm; Figure 6c) after the viewing and walking activities respectively. Compared with the bamboo forest environment, the subjects’ heart rates in the urban (CS) environment showed an obvious upward trend. After viewing and walking activities in the bamboo forest sites, the subjects showed a high level of peripheral oxygen saturation. Viewing activities in the bamboo forest environments were significantly correlated with the increase in peripheral oxygen saturation, especially in the YA site (97.47 ± 0.10%; *p < 0.05*; Figure 6d).The average peripheral oxygen saturation in the bamboo forests reached 97.5% after the viewing activities; YB site (97.37 ± 0.12%; Figure 6d) and DJY site (97.70 ± 0.12%; Figure 6d).The average peripheral oxygen saturation was 97.45% after the walking activities, which was 1.45% higher than that in the urban (CS) environment (96.11 ± 0.13%; Figure 6d).

### 3.3. Three Days of Bamboo Forest Therapy Was More Effective for Females

The three kinds of bamboo forest environments had a positive impact on the systolic blood pressure, heart rate, and peripheral oxygen saturation in male and female subjects (Figure 7). Compared to the urban (CS) environment, the male and female subjects in the bamboo forest environment had decreased physiological indexes, including systolic blood pressure, heart rate metrics, and the reported bamboo forest environment experience. However, the female subjects experience a lower systolic blood pressure and slower heart rate compared to the male subjects. The mean systolic blood pressure decreased by (4.12 ± 1.01 mmHg; Figure 7a) at YA site, (7.05 ± 1.09 mmHg; Figure 7a) at YB site, and (4.73 ± 0.75 mmHg; *p <* 0.05; Figure 7a) at DJY site, and the heart rate decreased by (4.15 ± 1.01 bpm; Figure 7c) at YA site, (7.30 ± 1.50 bpm; *p <* 0.05; Figure 7c) at YB site, and (4.92 ± 0.82 bpm; Figure 7c) at DJY site, respectively. Compared with the urban environment, the bamboo forest environment was more conducive to the increase and maintenance of peripheral oxygen saturation in male and female subjects. The results show that the bamboo forest environment can effectively do well for the physiological indexes on the human body, the results were even better for female subjects.

## 4. Discussion

Ulrich [37] pointed out that viewing in the natural environment with color slides can relax people’s heart rate, which can relieve stress and negative emotions. When people view the natural landscape, the effect of the plant landscape on human health can be inferred from the detection of changes in physiological indicators. Following the three-day bamboo forest therapy, university students showed a significant difference in their environmental evaluation (SDM) relative to those at the CS site. In terms of sensory perception, atmosphere, climate, place, and space, the subjects reported a better environmental experience in the bamboo forests. A beneficial environment such as a forest, which should include diverse vegetation and ecological components, helps improve the effect of physiological therapeutic interventions [38]. Studies have shown that monoculture forests also have therapeutic effects. A Korean study aimed at suggesting the reasonable guideline tree density of four coniferous tree species, Pinus *koraiensis*, Chamaecyparis *obtusa*, Pinus *densiflora*, Larix *kaempferi*, which improved the benefits of forest therapy [39]. Therefore, absolute illumination, temperature, and noise might be the environmental factors in the forest most conducive to human physiological relaxation. The two types of experimental sites in this study exhibited completely different environmental features, including diversity in visual, olfactory, tactile, and auditory stimulation. Previous studies have also concluded that forest therapy made subjects report feeling significantly more “comfortable,” “relaxed,” and “natural” according to the semantic differential (SDM) [28,40]. Because the urban environment was significantly different from the bamboo forest, it was unnatural, unhealthy, uncomfortable, too bright, and noisy [41]. In this study, the absolute illumination at the YB site (3774.65 ± 583.00 Lx; Table 2) was significantly higher than that at the YA site (877.09 ± 337.00 Lx; Table 2) and DJY site (815.15± 267.00 Lx; Table 2), and it was significantly lower than that at the CS site (6585.68 ± 881.00Lx; *p < 0.05*; Table 2). The temperatures at the YB (23.19 ± 0.50 °C; Table 2) and YA sites (26.90 ± 0.80 °C; Table 2) were close to 25 °C, which is a more suitable temperature for humans than the temperatures at the DJY site (18.71 ± 0.50 °C; Table 2) and CS site (28.87 ± 0.26 °C; *p* < 0.05; Table 2). The noise level at the YB site (36.75 ± 0.70 dB; Table 2) was significantly lower than that at the YA site (51.69 ± 0.90 dB; Table 2), DJY site (53.35 ± 0.50 dB; Table 2), and CS site (70.13 ± 0.68 dB, *p < 0.05*; Table 2). Based on our field investigation, the environments at the YA and DJY sites were significantly different from those at the YB site in terms of absolute illumination, temperature, and noise. Compared with the three other experimental sites, the environment at the YB site was more natural, comfortable, open, and bright with pleasing ambient noise; therefore, among the four sites, the YB site was the best for physiological relaxation among university students. Our research results showed that compared with urban (CS) sites, bamboo forest sites may be more conducive to decreased blood pressure and heart rate, and the viewing and walking activities had a positive impact on university students’ physiological indexes. The blood pressure values of the subjects at the bamboo forest site were less than those of the subjects at the urban site. Studies on the effects of stress and the cardiovascular system indicated that the forest environment significantly increased parasympathetic nervous activity and significantly suppressed the sympathetic activity of participants compared with the urban environment [42]. The changes in physiological indicators in the research results are completely in line with Ulrich [43], that is, the recovery was faster and more complete when subjects were exposed to natural rather than urban environments and the findings raised the possibility that responses to nature had a salient parasympathetic nervous system component. Li Q [44] conducted a study in Japanese people with two-hours walking in the forest, showing that habitual walking in forest environments may lower blood pressure by reducing sympathetic nerve activity and have beneficial effects on blood adiponectin. One of the studies in South Korea [26] showed similar results that in terms of the impact of 35 forest environments on human physiological relaxation demonstrated in the research results. A study with a dozen male college students who viewed and walked through forests confirmed that the forest environment favored parasympathetic activity [45].

Our research results show that the bamboo forest environmental experience was conducive to decreasing the female subjects’ systolic blood pressure and heart rate and to maintaining peripheral oxygen saturation at a higher level. Studies of individual differences in the physiological relaxation effects of forest therapy mainly refer to different behavioral patterns, and few studies have analyzed differences between genders. It is known that great individual differences are observed in physiological data collected in research on stress and relaxation, but few methods have been proposed to elucidate this variability. Ochiai [40,46] assessed nine hypertensive males to demonstrate the effects of a forest therapy program, and the comparison showed that forest therapy decreased systolic blood pressure. Furthermore, 17 middle-aged females participated in a study with the same experimental design and locations the following year. Substantial physiological benefits of forest therapy were observed in middle-aged females. According to the SD questionnaires, middle-aged females felt more “comfortable,” “natural,” and “relaxed” after forest therapy. Chorong Song [47] conducted a study that included 65 women in six forests and six city areas. Compared with viewing city areas for 15 min, viewing forest landscapes was associated with significantly higher parasympathetic nervous system activity (PNS) and lower sympathetic nervous activity and heart rate. Studies [32] have shown that when viewing and walking in bamboo forest sites, the heart rate mainly presents a downward trend. Between the two activities, the viewing activity had a more obvious effect on the subjects’ heart rate reduction. Hassan [34] studied the physiological effects of viewing a picture of a bamboo plant and found significantly lower systolic and diastolic blood pressure. Tsursunetsugu [48] investigated the physiological and psychological effects of viewing urban forest landscapes on 48 young male urban residents, and the results showed that the heart rate was significantly lower during every minute of viewing the forested areas than during the time spent viewing urban areas. Other studies have also shown that the mean heart rate was significantly lower when participants viewed a forest area than when they viewed an urban area [49,50]. Heart rate is an important indicator of the health of the cardiovascular system. This is especially true when people deal with stressful situations or are in stressful environments. Generally, an increase in heart rate is related to increased sympathetic excitation, while a decrease in heart rate is related to stimulation of the parasympathetic nervous system, which is related to stress reduction. Our study found that participants’ heart rates decreased when they participated in viewing and walking activities in the bamboo forests. Between the two types of activities, the viewing activity had a more obvious effect on decreasing the subjects’ heart rate. The observed increase in peripheral oxygen saturation may be conducive to the emotional calming effect. The viewing and walking activities in the bamboo forest sites were beneficial to improving peripheral oxygen saturation and maintaining peripheral oxygen saturation at a high level. The participants’ peripheral oxygen saturation was decreased at the CS site, while after the viewing and walking activities in the bamboo forest sites, the participants’ peripheral oxygen saturation was maintained at a higher level, and the participants tended to be calm and relaxed. This indicates that the bamboo forest environment was more conducive to the improvement in the oxygen concentration of the participants, possibly because the bamboo forest environment has a higher negative oxygen ion concentration. Some studies have shown that exposure to environmental negative oxygen ions improves performance efficiency and mental state [51,52], increases the oxygen content of blood, and decreases heart rate [53]. All these results are broadly in line with our finding that a sufficient oxygen-rich environment in the bamboo forest was conducive to improving participant health. 

## 5. Limitations

In this study, we obtained the changes in physiological indicators of the human body following different activities (viewing and walking) in both sexes (male and female) and in different ornamental bamboo forest environments. The effects of environmental factors on human physiology were quantitatively analyzed. However, this study also has the following shortcomings. First, due to time constraints, this study was only carried out in the summer, and future research is needed test the impact of the ornamental bamboo forest environment on human physiology in different seasons, and a comparative study of bamboo forests and other types of forests will help clarify the benefits of bamboo forest therapy. Second, the subjects in the study were all university students from Sichuan Agricultural University. The population as a whole has a high level of education and a good standard of living and health, which is not universal; therefore, the results can only represent the situation of some young people in the city. In the future, subjects should include people of different occupations, different age groups, and different life backgrounds. This would make the results more generalizable. Finally, this study evaluated the impact of the environment on human physiology only. In the future, more indicators that can reflect the relationship between the environment and human health, such as physiological, psychological, and endocrine hormones, should be included in the study for a more comprehensive reflection of the environmental impact on physical and mental health.

## 6. Conclusions

Physiological data from this field experiment provide important scientific evidence on the health benefits of bamboo forest therapy. The results support the concept that forest therapy has positive effects on physical and mental health, indicating that it can be effective for health promotion. There were significant differences in the subjects’ subjective environmental evaluation (SDM) of bamboo forests and cities. In terms of sensory perception, atmosphere, climate, place, and space, the subjects obtained a better environmental experience in bamboo forests. Viewing and walking activities in the bamboo forest environment were beneficial, as indicated by the positive changes in human physiological indicators, in terms of effectively reducing blood pressure, slowing the heart rate, and maintaining peripheral oxygen saturation at a high level. Bamboo forest therapy can relieve physical stress and stabilize emotions. Relative to the walking activity, the viewing activity had a more obvious effect on the subjects’ heart rate slowing, and the bamboo forest therapy session had positive effects on the systolic blood pressure, heart rate, and peripheral oxygen saturation of male and female subjects. The bamboo forest environmental experience was more conducive to the decrease in female subjects’ systolic blood pressure and heart rate, and it maintained peripheral oxygen saturation at a higher level. This indicates that the bamboo forest environment can effectively relieve the pressure on the human body, relieve emotions, and increase energy. The three-day bamboo forest session effectively improved the short-term psychological well-being of female participants.

## Figures and Tables

**Figure 1 ijerph-17-03238-f001:**
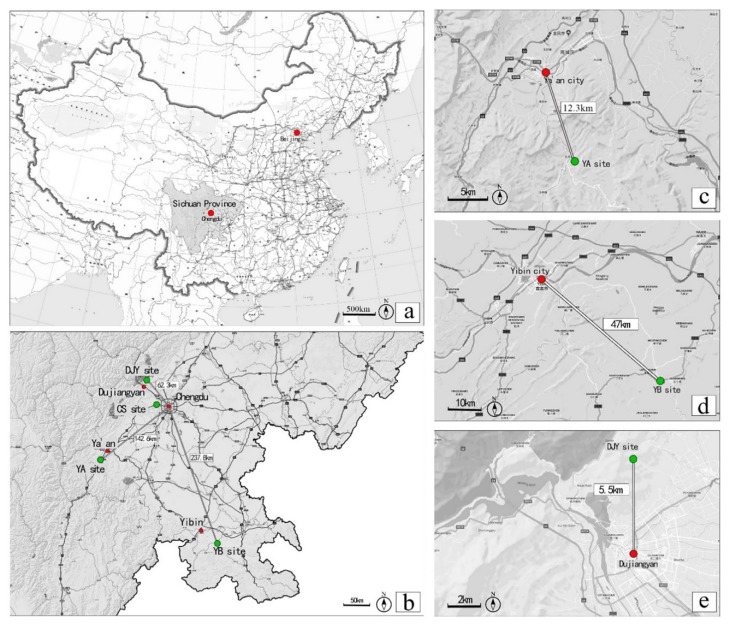
Maps of experimental sites. (**a**) Sichuan Province in China; and (**b**) the four sites in Sichuan Province; (**c**) Ya’an sites (YA); (**d**) Yibin sites (YB); (**e**) Dujiangyan sites (DJY). The red points mark the downtown areas, and the green points mark the location of each site.

**Figure 2 ijerph-17-03238-f002:**
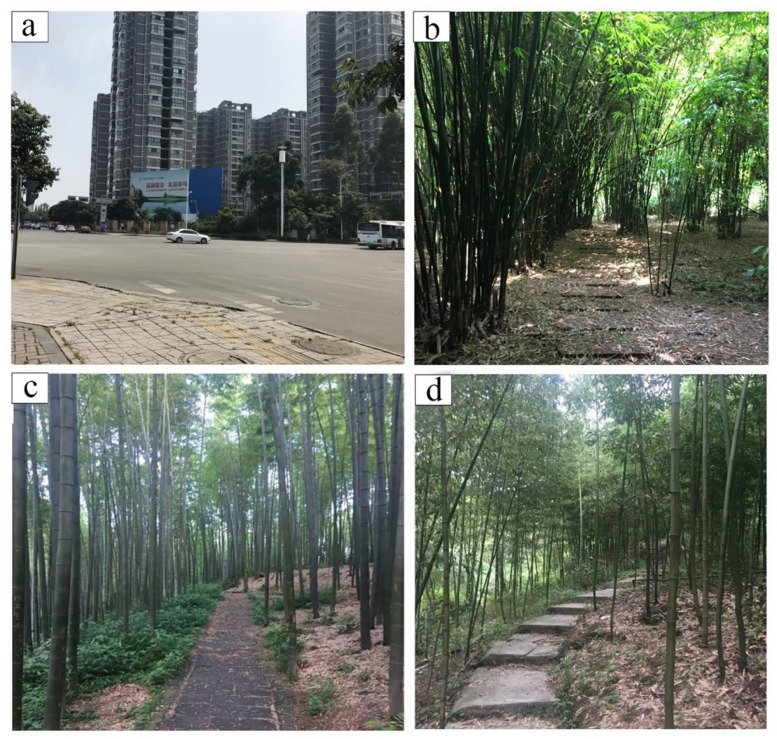
Photographs of the studied sites. (**a**) The CS site was located at a crossroad in a typical urban environment with cars, buildings, markets, hotels, and companies; (**b**)–(**d**) show the bamboo forest sites with path. (**b**) *Neosinocalamus affinis* bamboo forest; (**c**) *Phyllostachys heterocycla* bamboo forest; (**d**) *P. praecox f. prevernalis* bamboo forest.

**Figure 3 ijerph-17-03238-f003:**
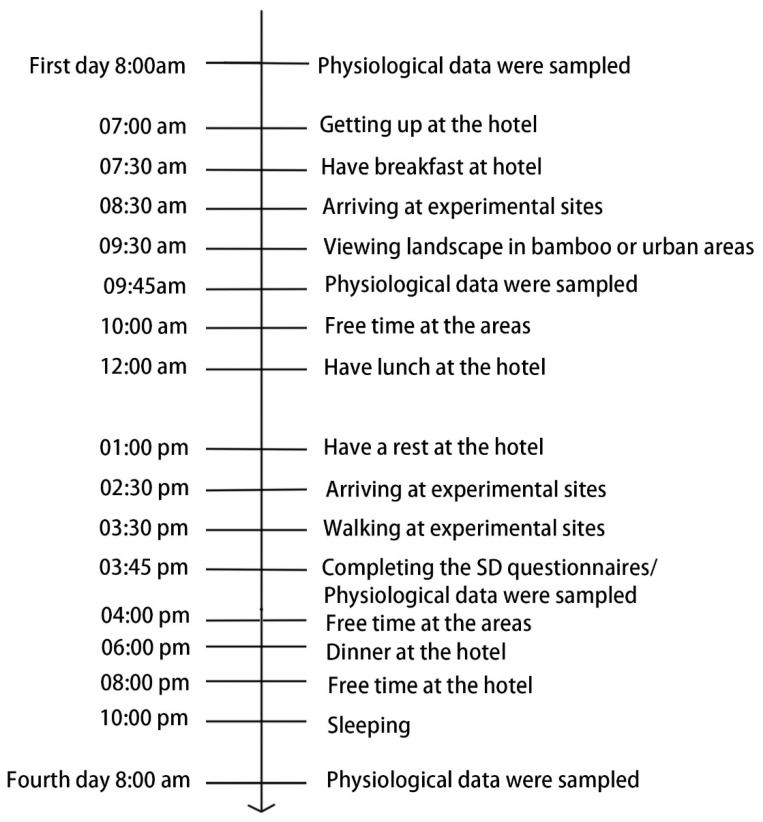
The itinerary for the subjects exposed to the bamboo forests or the urban environment.

**Figure 4 ijerph-17-03238-f004:**
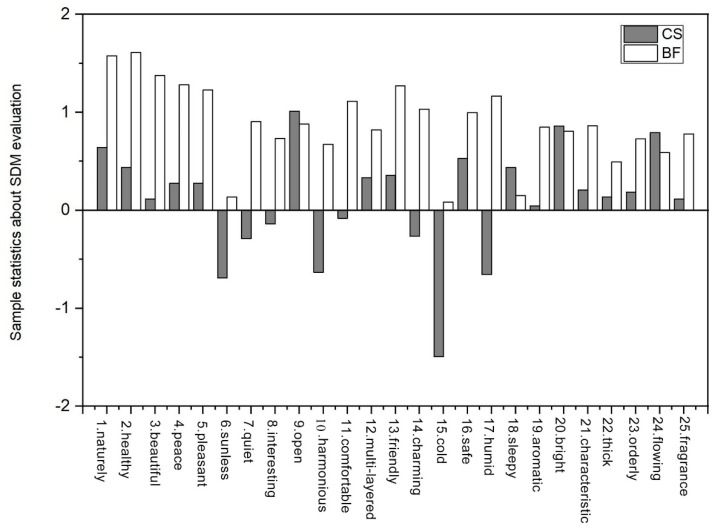
Comparison of environmental characteristics between the bamboo forests and the urban environment.

**Figure 5 ijerph-17-03238-f005:**
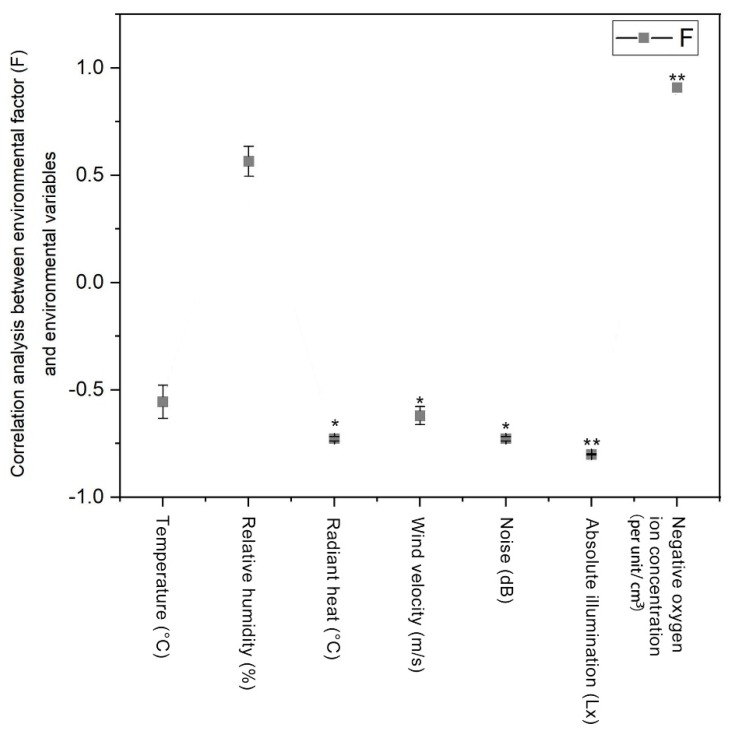
The spatial and sensory factors identified by the SDM as they relate to the seven environmental factors. Data are presented as means ± SD; *n* = 348; * *p* < 0.05; ** *p* < 0.01; integrated environmental assessment factor score (F) = [sensory factor score (F1) x factor variance percentage senses + atmosphere factor score (F2) x atmosphere factor variance percentage + climate factor score (F3) x climate factor variance percentage + location factor score (F4) x place factor variance percentage + space factor score (F5) x space factor variance percentage]/percentage of the total variance, is obtained by computing the final composite scores of each environment factor.

**Figure 6 ijerph-17-03238-f006:**
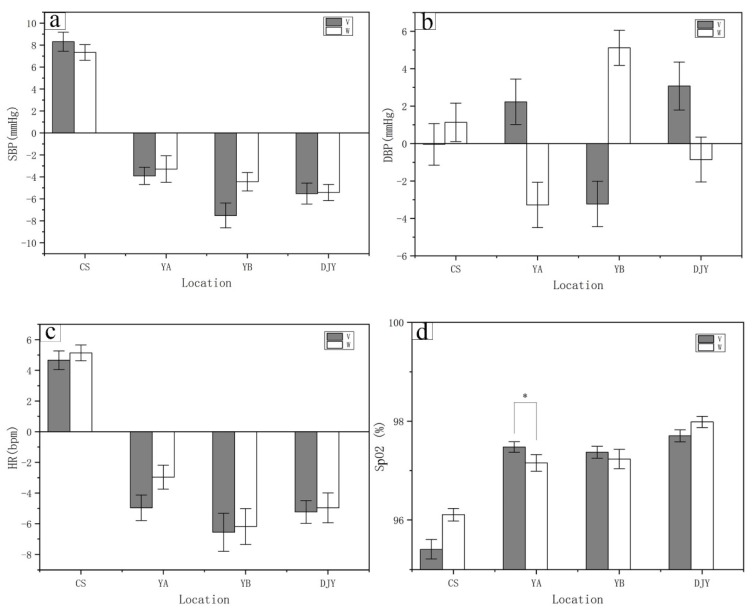
Systolic blood pressure, diastolic blood pressure, heart rate, and peripheral oxygen saturation in participants after bamboo forest and urban program. A paired *t*-test was used to compare the data between viewing and walking activities after three days of bamboo forest therapy. The data are presented as the mean ± SEs; * *p* < 0.05; significant difference between the data for viewing and walking activities was observed for the physiological indexes by paired t test. CS (*n* = 30), City site; YA (*n* = 30), Ya’an sites. YB (*n* = 30), Yibin sites. DJY (*n* = 30), Dujiangyan sites. (**a**) SBP, systolic blood pressure; (**b**) DBP, diastolic blood pressure; (**c**) HR, heart rate; (**d**) SpO2, peripheral oxygen saturation.

**Figure 7 ijerph-17-03238-f007:**
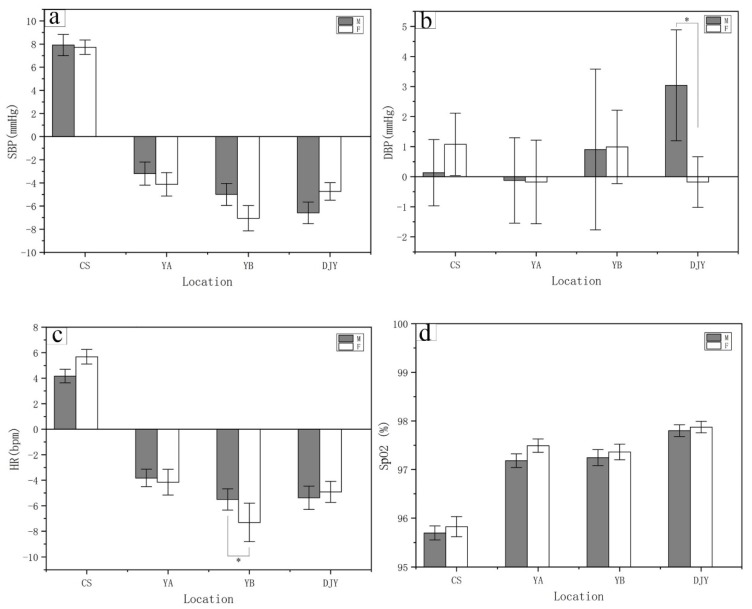
Changes in systolic blood pressure, diastolic blood pressure, heart rate, and peripheral oxygen saturation in participants after bamboo forest and urban program. One-way ANOVA (Turkey’s test) was used to compare the data between male and female subjects after three days of bamboo forest therapy. The data are presented as the mean ± SEs; * *p* < 0.05; significant difference between the data for male and female subjects for the physiological indexes by paired t test. CS (*n* = 30), City site; YA (*n* = 30), Ya’an sites. YB (*n* = 30), Yibin sites. DJY (*n* = 30), Dujiangyan sites. (**a**) SBP, systolic blood pressure; (**b**) DBP, diastolic blood pressure; (**c**) HR, heart rate; (**d**) SpO2, peripheral oxygen saturation.

**Table 1 ijerph-17-03238-t001:** Basic information of the sampling subjects at the four sites included in this study.

Parameter	CS		YA		YB		DJY	
	Male	Female	Male	Female	Male	Female	Male	Female
Sample size (count)	15	15	15	15	15	15	15	15
Age (years)	21.6 ± 0.3	21.3 ± 0.3	22.1 ± 0.4	21.4 ± 0.3	21.5 ± 0.4	20.8 ± 0.4	21.8 ± 0.5	21.2 ± 0.3
Weight (kg)	65.5 ± 1.2	51.8 ± 1.6	63.9 ± 1.3	49.7 ± 1.5	64.4 ± 1.3	47.1 ± 0.9	63.9 ± 1.7	50.4 ± 1.7
Height (cm)	175.2 ± 0.5	164.5 ± 1.0	175.6 ± 0.6	164.0 ± 1.1	174.9 ± 0.8	162.5 ± 1.3	173.4 ± 1.0	163.3 ± 1.0
BMI (kg m−2)	21.3 ± 0.5	19.1 ± 0.4	20.7 ± 0.5	18.5 ± 0.4	21.0 ± 0.3	17.8 ± 0.2	21.2 ± 0.5	18.9 ± 0.7
SBP (mmHg)	110.0 ± 2.8	96.0 ± 2.0	109.6 ± 2.2	99.8 ± 2.4	103.8 ± 2.9	97.6 ± 3.6	103.7 ± 3.0	100.2 ± 2.7
DBP (mmHg)	59.9 ± 1.6	57.6 ± 2.2	58.9 ± 1.3	60.0 ± 1.9	63.1 ± 2.0	63.3 ± 2.4	61.9 ± 1.5	63.6 ± 1.5
HR (bpm)	63.6 ± 2.7	64.8 ± 2.4	63.8 ± 2.8	65.2 ± 3.2	64.6 ± 3.0	65.8 ± 2.9	66.6 ± 2.9	67.9 ± 2.0
SpO_2_ (%)	97.1 ± 0.3	97.3 ± 0.3	97.8 ± 0.3	97.3 ± 0.4	96.8 ± 0.5	97.4 ± 0.4	98.1 ± 0.2	97.0 ± 0.4

Body mass index (BMI) = weight (kg)/[height (m)^2^]. All data are presented as the mean ± SEs (*n* = 15 per group). CS: city site; YA: Ya’an site; YB: Yibin site; DJY: Dujiangyan site. DBP: diastolic blood pressure; SBP: systolic blood pressure; SpO_2_: blood oxygen saturation level.

**Table 2 ijerph-17-03238-t002:** Comparison of the environmental factors of the four environmental sites.

Parameter	CS	YA	YB	DJY
Temperature (°C)	28.9 ± 0.26	26.9 ± 0.8	23.2 ± 0.5 *	18.7 ± 0.5 **
Relative humidity (%)	60.5 ± 2.53 *	79.0 ± 3.8	74.5 ± 4.0	87.5 ± 3.6
Radiant heat (°C)	34.5 ± 0.73	26.3 ± 0.3 *	26.2 ± 1.0 *	17.8 ± 0.3 **
Noise (dB)	70.1 ± 0.68	51.7 ± 0.9 *	36.8 ± 0.7 **	53.4 ± 0.5 *
Absolute illumination (lx)	6585.7 ± 881	877.1 ± 337.0 **	3774.7 ± 583.0 *	815.2 ± 267.0 **
Wind velocity (m s−1)	0.9 ± 0.19a	0.1 ± 0.1b *	0.2 ± 0.1b *	0.4 ± 0.1 *
Negative oxygen ion concentration (per unit/cm^3^)	573.33 ± 15.08	1273.89 ± 45.37	718.33 ± 31.52	895.56 ± 40.99

Data are presented as the mean ± SE; * (*p* < 0.05); ** (*p* < 0.01) CS: city site; YA: Ya’an site; YB: Yibin site; DJY: Dujiangyan site.

**Table 3 ijerph-17-03238-t003:** Results of principal component analysis. Data show the results of principal component factoring based on the results obtained from the SDM questionnaire, data are presented as the means ± SD; *n* = 348; *p* < 0.05.

Name	Evaluation Item	Factor Load				
		Factor NO.1	Factor NO.2	Factor NO.3	Factor NO.4	Factor NO.5
**Sensory factors**	1.naturely	0.75	−0.01	0.06	−0.01	0.18
	2.healthy	0.81	0.10	0.11	0.00	0.08
	3.beautiful	0.84	0.08	0.14	0.03	0.14
	4.peace	0.81	0.23	0.05	0.08	−0.10
	5.pleasant	0.81	0.20	−0.02	0.07	0.01
	6.sunless	0.54	0.36	0.17	−0.03	0.13
	7.quiet	0.64	0.13	0.05	0.28	0.04
	8.interesting	0.75	0.28	0.06	0.06	0.20
	9.open	0.67	0.27	0.12	0.09	0.11
	10.harmonious	0.69	0.30	0.21	0.07	0.16
**Atmosphere factors**	11.comfortable	0.48	0.50	0.38	−0.23	0.12
	12.multi-layered	0.45	0.54	0.07	0.19	−0.10
	13.friendly	0.32	0.72	0.15	−0.06	0.23
	14.charming	0.09	0.61	−0.47	0.26	0.14
	15.cold	0.35	0.40	0.01	0.38	0.09
	16.safe	0.30	0.69	0.16	−0.03	0.18
**Climate factors**	17.humid	0.09	0.18	0.68	−0.24	0.17
	18.sleepy	0.28	0.00	0.71	0.12	0.05
	19.aromatic	0.52	0.09	0.53	−0.09	0.21
**Place** **factors**	20.bright	−0.22	0.39	0.41	0.41	−0.22
	21.characteristic	0.16	0.07	−0.18	0.68	0.08
	22.thick	0.01	−0.03	0.00	0.77	0.19
**Space** **factors**	23.orderly	0.31	−0.07	0.12	0.31	0.63
	24.flowing	0.34	0.29	−0.14	0.21	0.53
	25.fragrance	−0.06	0.27	0.22	0.03	0.69

**Table 4 ijerph-17-03238-t004:** Comprehensive evaluation score of environmental factor (F), comprehensive evaluation score of environmental factor. According to the factors within the contribution of different evaluation index for each environment factor, calculate the final score of different factor = the sum of (environment evaluation index by correspondence factor load)/(the sum of corresponding factor load).

Location	F1	F2	F3	F4	F5	F
**CS**	0.15	0.23	−0.98	0.71	0.22	0.09
**YA**	1.26	0.68	0.16	0.57	0.82	0.96
**YB**	1.29	1.01	0.20	0.30	0.87	1.01
**DJY**	1.18	0.70	0.41	0.54	0.55	0.92

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
