# Peer review of "Benefits of a Three-Day Bamboo Forest Therapy Session on the Physiological Responses of University Students"

_ijerph, 2020, doi:10.3390/ijerph17093238_

Round 1

Reviewer 1 Report

This paper surveyed the effects of bamboo environment.

There are many bamboo stands in China and Asian countries.

Therefore,  the results of this experiment certainly show the possibility of human health promotin by bamboo stands!

However, both of the experiment design and method are stereo-type of past experiments of forest therapy.

If you try to create new experiment design and method, you can find new undiscovered effects, too!

Reviewer 2 Report

Here are my detailed comments to what has potential to be an interesting and valuable paper:

Lines 42-45 Reword the first sentence to reflect on the fact that while urban environments have many positive aspects they also have negative ones, such as impact son health and wellbeing – it is clumsy as written.

Line 47 It is not just preference – people are forced economically to move to urban areas – we would not say that large slums and barrios are places people prefer to live.

Lines 53-60 You focus on college students here and you use them as research subjects – so also saying that results from them are reliable. This needs to be reconsidered in the light of the aims of the paper – is it to test bamboo forest therapy to see if it works like regular forest therapy or to test how bamboo forest therapy works for college students. If the former, then you don’t need to point out stressful conditions of students – you are using them as test subjects just like many other studies – including of forest bathing. I don’t see how you can do both in this paper. You need to decide.

Line 60 on – break the paragraph here into a new one – this is far too long and it includes several different issues under discussion. Do not keep saying “researchers have…” at the beginning of so many sentences. It is repetitive.

Line 91 – break the paragraph here as the subject switches to bamboo forests from discussions of forest therapy. This may need to be expanded to describe the qualities of bamboo forests – and maybe to use the definition of the FAO that bamboo is counted as a forest type and not just trees.

Line 98 – the research objective, hypothesis and questions should be expanded and clearly stated here, better than they are, with the clarification of what you are testing as per my notes above.

Section 2.1 – you chose four sites why and how were they chosen? Clarify. Fig 1b has no scale – how far away are these from Chengdu? How easy would it be for residents of Chengdu who might enjoy forest therapy to travel to them? What is the relevance of the different bamboo forests? Did you hypothesise that different species might have different effects? What is the relevance of the fact that Ya’an is the city of giant pandas?

Fig 2 – it is difficult to see the scale of the forest – is site C a bigger species of bamboo? If you replaced the pictures with ones with people it would help to understand the atmosphere and relative scale. Two show a path but C does not.

Line 125 You don’t need to say they were college students at the university – just students at the university. The term college students is very American usage. What did they study? Would this have made a difference if they were eg forestry students?

Section 2.2. This is confusing – is it the case that of the four groups three went to one each of the forest sites and stayed there and only did the study there in the one place? I cannot find anything about where the 4th group went – did they stay in the city as a control? Did they also stay in a hotel? What did they do? This is mentioned in the timetable but nowhere else. The city was hotter and less comfortable than the forests – did they have shade? More clarity of explanation is needed. At present the methodology does not allow exact replication.

Lines 181-182 What is in incorrect questionnaire? How were any invalid since you had the students there under controlled conditions?

Lines 185-189. Measuring BP etc takes time – was the order of taking it changed each time to take into account changes from sitting waiting? Would the wait have an impact on the results?

Fig 4 – is a line graph the correct way of showing the results when each are categorical variables? – it looks like trend lines over time when in fact it is not. A bar chart would be much clearer for presenting these results. The caption should use the same terms as the section title and the measurement name.

Line 244 – finish the sentence with “…when compared to the urban setting” or similar.

Figures 6 and 7 – why is the urban setting named CK when everywhere else it is CS? This is confusing. What do the letters a and b refer to? I cannot find anything explaining them in the text.

You took measurements of environmental conditions in each site yet you do not carry out ant analysis to see if these had any effect or correlation – could it have been the temperature and shade that accounted for the improvements? Why collect the data if you don’t do anything with it? You could correlate the perception of climate with the measurements (perceived thermal comfort with objective temperature and humidity for example).

Line 287 You refer to Kim about forests having diversity – yet the bamboo forests are extremely monotonous – a key issue that makes them different and may account for some differences yet you don’t discuss this.

Line 296 on – here you refer to the environmental conditions in the discussion without having the data analysis to back up the assertions. Please carry out some to support these kinds of statements.

Line 315-316. You refer to Ulrich’s SRT here but for some reason you did not include it in the theoretical part of the lit review – an important omission.

Lines 358-360 – you talk about an oxygen rich environment of the forest and make claims about its effect yet you carried out no measurements of oxygen concentrations as far as I can ascertain. So can you be as certain as this in your assertion?

Limitations – you did not test the bamboo forest against a “normal” forest of trees as well as the urban setting – this would have tested the hypothesis more strongly.

Conclusions – if you had more formal research questions you would have clearer statement of the conclusions, although they are OK

General – the language is OK. I am surprised that there is no reference to any work by Qing Li who is the definitive author on what is now called “forest medicine”.

Reviewer 3 Report

This manuscript potentially adds to the literature on the health benefits of restorative trips to forests. It suggests that bamboo forests, as well as coniferous forests may be restorative. The study has a solid design, but the manuscript could be improved to clarify the results and their interpretation.

General comments:

The title should refer specifically to cardiovascular responses. "Physiological" is too broad.

Please check the citation format. Some in-text citations contain the initials of author’s given names. Also, the reference list uses numbered references, but authors’ last names are used in the in-text citations. Please, use either reference numbers in the in-text citations or list the references in alphabetical order by first author in the reference list. The current presentation makes it difficult to find references.

Statistical Methods.

Please consider reporting the confidence intervals for your groups instead of standard deviation or standard errors. See: Harrington, D., R. B. D’Agostino, C. Gatsonis, J. W. Hogan, D. J. Hunter, S.-L. T. Normand, J. M. Drazen and M. B. Hamel (2019). "New Guidelines for Statistical Reporting in the Journal." New England Journal of Medicine 381(3): 285-286. https://www.nejm.org/doi/full/10.1056/NEJMe1906559 and Amrhein, V., S. Greenland and B. McShane (2019). "Scientists rise up against statistical significance." Nature 567(7748): 305-307. https://www.ncbi.nlm.nih.gov/pubmed/30894741

Where means and variances are included, the figure legends and table captions should clearly state whether the variance measures are standard deviations or standard errors, better yet, use 95% confidence intervals.

Specific comments:

Introduction: The primary value of this paper is the fact that the study examines the effects of exposure to bamboo forests vs urban areas, whereas much of the rest of the research literature emphasizes exposure to coniferous forests. The introduction could be improved by including more background on the types of forests that have been studied and describing why the use of bamboo forests is important. The authors touch on the role of bamboo forests at the end of page 2, but do not explain the importance of this study relative to other reports in the literature. See section 2 "Exposure Science" in Frumkin, H., G. N. Bratman, S. J. Breslow, B. Cochran, P. H. Kahn, Jr., J. J. Lawler, P. S. Levin, P. S. Tandon, U. Varanasi, K. L. Wolf and S. A. Wood (2017). "Nature Contact and Human Health: A Research Agenda." Environ Health Perspect 125(7): 075001. https://www.ncbi.nlm.nih.gov/pubmed/28796634

Page 2, lines 46 to 95 and continuing on to page 3. This paragraph covers too many topics. Please divide it into separate paragraphs discussing urbanization, research using college students, background on research on the health benefits of nature, the choice of outcomes variables, and the choice of forest types.

Page 2, line 59. I believe the citation to “Kanplan 1989” is a reference to a paper by Stephen Kaplan from 1989. Please, correct here and in reference list.

Page 2, line 60. “… most scientific and representative.” Representative of what? Please, provide a description of this comparison. Also, the Kaplan reference is from 1989. There has been much more research on a variety of different populations. In 2020 it does not seem appropriate to cite a paper from 1989 to justify the use of college students as a population representative of all other demographic groups.  See section 4 "Diversity and Equity" in Frumkin et al. 2017 (reference given above).

Page 2, line 78 “find” is a poor word choice. Do you mean “conduct”?

Page 2, line 82. “cerebra” should be “cerebral”

Page 3 Figure 1. Text on the maps is too small and cannot be read.  Also, please give the latitude and longitude coordinates for each study site either in the text or the figure legend.

P 5 Lines 145-148. Please provide additional information on and references to the methods used to measure each of the environmental variables.The authors highlight the differences between the city and the forest sites, but gloss over differences among the forest sites.  Note: No mention of a measurement of oxygen content of the air was made; reference is made to differences in oxygen content of the air later in the manuscript. Either these measurements need to be provided or the discussion of the oxygen concentration of the air that appears later in the manuscript must be removed.

Page 5 Table 2. Is “Radiant heat” air temperature? “Radiant Heat” is not synonymous with ambient air temperature. Please, check the definitions. If radiant heat was in fact measured, please describe the method used and the rationale for using radiant heat instead of ambient air temperature. Negative air ion data are not included in the table although the text says they were measured. Please, either include the data in the table or delete the comments about having measured negative air ions. In the footnote, the description of letters a, b, and c is confusing. Is it correct to say that values marked with the same letter did not differ significantly from each other?

Page 5 lines 155-158. Where was the hotel in the city located relative to the study site?  What is meant by “excess emotions”? This phrasing seems arbitrary and vague.

Page 6 line 166. Need to include the fact that a “nap time” followed lunch as shown on the figure. Did participants actually sleep (“nap”) or were they simply resting quietly, but awake? If they did not sleep, it is inappropriate to call this nap time.

Page 6 line 171. There is only one index (not indices) and it is the semantic differential scale, which is a measure of attitudes. The semantic differential scale measures peoples opinions of things, it does not measure a person's psychological state.

Page 6 lines 171-183. The description of the semantic differential test is confusing. Please, simplify to explain that the test provides a measure of a person’s attitude toward things. Also, include a reference describing the development and use of semantic differential scales.

Page 6 Line 184. Please use “cardiovascular measures” rather than “physiological”. Physiological is too broad.

Page 7 lines 185-189. More detail needs to be given regarding the timing of when the cardiovascular measures were made. First, is an “intervention” a 15 minute session of sitting or walking? When were measurements made relative to the beginning and end of the sitting and walking sessions? The lag time between when people finish a session, especially a walking session, will greatly influence all of the measures made simply due to changes in posture and activity levels.  Also, one measure of recovery is how long it takes heart rate to return to baseline after a period of activity.

Page 7 lines 191 – 203. Why were sex differences assessed using t-tests? Using a two-factor ANOVA with sex and location as the treatment variables would allow assessment of whether males and females respond to the interventions differently. Also reference is made later in the manuscript to the impact of the “3 day” visit, however no comparison is made to assess the effect of time (day). Did the response on day 1 differ from the response on day 3? Was it appropriate to average across all three days? It seems that a repeated measures ANOVA should be conducted.

Page 7 lines 195-196. “viewing and walking variables”. This sentence is unclear. The outcome variables are cardiovascular measurements. These measurements were taken before and after viewing and walking sessions, viewing and walking are not outcome variables.

Page 7 lines 198-199. Again, how were possible differences in the timing of measurements accounted for in the calculation of change? How were the pre- and post-intervention times standardized?

Page 7 through page 8. Description of the results from the semantic differential test. Either the contribution of the detailed analysis of the SD needs to be reflected in the title and the introduction of the paper or this section needs to be reduced in scope. The presentation seems unbalanced. Also, given the differences observed among the forest sites (Table 2) it seems inappropriate to average the SD scores from the 3 forest sites without demonstrating there that this compilation of the data is justified.  

Page 7 Figure 4. Viewing these data as continuous curves seems in appropriate. The descriptors are not continuous variables. Also, without point markers to identify specific measures, it is impossible to figure out what value on the line aligns with each descriptor.

Page 8 Figure 5. This is a table not a figure. It is also impossible to read. The table needs to be explained. The reader is left to assume that this is a comparison of the contrasts for SD scores between forest and urban sites, that needs to be described in the labels and the caption. Also, given the large number of comparisons, was the assessment of significance corrected for false positives?

Figures 6 and 7, Please expand the figure legends to explain what the letters associated with the bars are and describe whether the error bars represent standard deviations, standard errors, or confidence intervals. Axes of both figures should be labeled as “change in…”

Figure 6. The title describes “changes after viewing or walking” but the data for the partial pressure of oxygen is the raw measure (%). The SPO2 panel can be redrawn breaking the ordinate (Y axis) to emphasize values between 90 and 100% saturation. The normal range for SP02 is 95-100. Including the range from 0 to 90% distorts the graph.

Figure 7 and all discussion of sex differences. Given that the reported sex differences in heart rate and blood pressure are typical sex differences in these measures, it is not clear that females responded any differently than males to the interventions. Only a two way ANOVA showing an interaction or additive effect of sex and location would demonstrate this.

Pages 8-9. Lines 243-264. The authors need to be careful in the use of the terms “decrease” versus “different”. They have conducted a difference in difference analysis. They also need to be transparent in reporting statistics. For example in lines 250-251 they describe a “downward trend” but do not report the F, p values or confidence intervals.

Page 10 Line 280. The paper by Ulrich (1981) did not assess peoples’ responses to a walk. The study was done by showing people pictures.

Page 10, Line 284. Cannot describe the results as occurring “following a three-day …” because no analysis of differences in day was conducted. The same results may have been the same if analyzed for the first day.

Page 11 lines 304-306. The results are not broken down by forest location, so it impossible for the reader to assess the validity of this statement.

Page 11 lines 322-326 and again in lines 335-340. Heart rate is controlled by both the parasympathetic and sympathetic nervous systems. A decrease in heart rate could reflect either an increase in parasympathetic or a decrease in sympathetic input to the heart. It is impossible to tell from the data provided which component of the autonomic nervous system was changed. Probably both. Heart rate variation gives a better indication of relative changes in parasympathetic and sympathetic variation.

Page 11 line 348. The authors should re-read Ulrich’s 1984 paper. The paper describes differences in pain medication use and length of hospital stays. The paper contains no data on heart rate or blood pressure. Ulrich, R. S. (1984). "View through a window may influence recovery from surgery." Science 224(4647): 420-421. http://www.ncbi.nlm.nih.gov/pubmed/6143402

Page 12 line 359-360. No measure of environmental oxygen content was made. The authors cannot state that the bamboo forests were more oxygen rich than the city. The observed differences in SPO2 could result from changes in blood flow.

Page 12 line 387. No measurements were made of perceived stress, only heart rate, blood pressure, and peripheral oxygen saturation, thus it is inappropriate to say that the bamboo forests reduced physical stress and stabilized emotions.

Page 12 line 388. Because changes in posture and physical activity can affect all of the cardiovascular measures used, without knowing the time delay between when participants stopped walking and when the post-intervention cardiovascular measurements were made, it is impossible to say that the viewing intervention had a greater effect than the walking interventions. Also, no statistical comparisons of the outcomes from the walking and viewing interventions were made.  

Reviewer 4 Report

The main objective of this study is to explore the physiological benefits of bamboo forest therapy on college students. By taking experiments on 120 students (60 male - 60 female) this study was conducted in three different bamboo forests and compared with an urban area in Chengdu - Sichuan Province, China. In addition, the design of this study also assessed differences in the effects of therapy if the respondents only watching or walking. The results show that three-day bamboo forest therapy was improved the physiological well-being of the college students and the viewing activities had a more effective on decreasing heart rate. Moreover, compared with the urban area, the bamboo forest sites effectively improved the physiological health, decreased the physical pressure, and stabilized the physiological indicators. This study might have the potential to be published. But the authors need to consider some comments below.

*general

  1. Need a slight correction for typos or technical writing
  2. It was known that several studies have discussed the benefits of three-day bamboo therapy. However, for the layman, the question that may arise is why should "three days"? Are indicators of physiological effects effectively identified during that time? I suggest that the authors can explain briefly and it would be better if add some references.

*introduction

  1. I identified some sentences quite similar to the previous study by Lyu et al. (2019) - https://doi.org/10.3390/ijerph16244991 (also found in the materials and methods section)
  2. The authors discussed a lot about the benefits of forest (in general) related to health and I did not find a part that discusses the benefits of bamboo forest related to health specifically. I suggest the authors can add related information.

*materials and methods

  1. Referring to location, why did this study choose experiments in three bamboo forests that generally have the same character? Why not consider comparing if the experiment was carried out in general forests (e.g. tree-forest)? Thus, the effects of nature on physiological conditions will be more varied, if located in an urban area, bamboo forest, or tree-forest.
  2. Figure 1. I suggest adding a legend for the symbols and please present a clear map (high resolution)
  3. Regarding the subjects selected, is it possible to consider the socio-economic background of respondents which might influence their perception of bamboo forests?
  4. Why did this study use different groups in different study locations? Why did not use the same groups to identify physiological effects at different locations - maybe it only requires time adjustments to do field experiments?

I think it makes sense to identify the different effects of different environmental conditions (urban site or bamboo forests) if the participating respondents are the same.

  1. Table 1 and Table 2, because of this study seem like the development from a previous study (Lyu et al., 2019). Please give citations related to the same data or information - provide an explanation that this study refers to previous research. Moreover, in Table 2 I did not find statistical data for 'negative air ionization'
  2. This study considers environmental factors in the analysis such as temperature, humidity, etc. (line 145 – 146). However, for specific conditions at urban sites, does this study consider the effects of air pollution that might affect the physiological conditions of the respondents?

*results and discussion

  1. Figure 4. I suggest adding axis information (X, Y) to the graphic
  2. Figure 6 and Figure 7, please describe clearly the information in the legend (V, W, and M, F) - at least mention in the discussion part about those symbols.
  3. In the discussion section, there is no clear references that discuss why female has a significant effect than male (line 335 – 337). Please add some references to support the results

Round 2

Reviewer 2 Report

This is a big improvement on  the previous version but still has a few issues to be addressed - in part I think because the new text and the old text (and pictures/graphs) are both still there. See my notes for various things I spotted when going over the paper:

Line 59 – split the paragraph here as the subject focus changes

Line 61 and 62 – what is subhealthsub health? (it seems you have made a bit of a mess of adding corrected text in places and leaving the old text – I see it further on too – or is it that you have  left the old text in and added the new in red? Still confusing!)

Line 63 – start referring to university students here and then use the term the rest of the way though the paper

Lines 76-82 – you repeat the text about Paracelsus

Line 89 – split the paragraph when introducing forest therapy and probably you should introduce it with reference to Li Qing here – as you say, the father of forest therapy.

Line 131 – “Ahmad Hassan [40] was conducted…” is poor grammar – please check this and other sentences where some things have been added.

Lines 136-8 – “However, to our knowledge, few reliable scientific data about the benefits of bamboo forest therapy on psychophysiological responsesphysiological recovery have not been investigated experimentallyobtained from field experiments” this is messy and needs to be rewritten.

Line 142 on – “Therefore, the aim of the present study was to investigate the physiological health benefits of three-days bamboo forest therapy. Physiological effects were compared between men and women when viewing and walking to provide scientific evidence for the physiological health benefits of bamboo therapy. This study will firstly discuss the results of the environmental evaluations in the sequence of the SDM, followed by the discussion of results of the relationship between environment factors to environmental evaluations. Lastly, the physiological benefits of bamboo forest therapy will be discussed” lists tasks, not research questions – so the aim and purpose is still not as clear as it could be. You refer to SDM here but SDSM in the abstract – check which it is.

Line 155 – you still say Ya’an is famous for pandas, an irrelevant reference as you agreed…

Line 204 – “Use bracket support in urban areas” is not a sentence and reads – like other parts of this new section – like an instruction manual. Please revise.

Line 293 – as a result of replacing the line graph with bar chart – much better by the way – you now leave text saying “contour curve” which now needs to be changed.

Line 472 – this big paragraph should be split – maybe when you talk about females’ responses.

Line 522 – be careful about how much confidence you show in Ulrich’s 1984 experiment – it is cited a lot but is not as clear-cut as you make it sound and more of a suggestion as to the implications.

Line 59 – split the paragraph here as the subject focus changes

Line 61 and 62 – what is subhealthsub health? (it seems you have made a bit of a mess of adding corrected text in places and leaving the old text – I see it further on too – or is it that you have  left the old text in and added the new in red? Still confusing!)

Line 63 – start referring to university students here and then use the term the rest of the way though the paper

Lines 76-82 – you repeat the text about Paracelsus

Line 89 – split the paragraph when introducing forest therapy and probably you should introduce it with reference to Li Qing here – as you say, the father of forest therapy.

Line 131 – “Ahmad Hassan [40] was conducted…” is poor grammar – please check this and other sentences where some things have been added.

Lines 136-8 – “However, to our knowledge, few reliable scientific data about the benefits of bamboo forest therapy on psychophysiological responsesphysiological recovery have not been investigated experimentallyobtained from field experiments” this is messy and needs to be rewritten.

Line 142 on – “Therefore, the aim of the present study was to investigate the physiological health benefits of three-days bamboo forest therapy. Physiological effects were compared between men and women when viewing and walking to provide scientific evidence for the physiological health benefits of bamboo therapy. This study will firstly discuss the results of the environmental evaluations in the sequence of the SDM, followed by the discussion of results of the relationship between environment factors to environmental evaluations. Lastly, the physiological benefits of bamboo forest therapy will be discussed” lists tasks, not research questions – so the aim and purpose is still not as clear as it could be. You refer to SDM here but SDSM in the abstract – check which it is.

Line 155 – you still say Ya’an is famous for pandas, an irrelevant reference as you agreed…

Line 204 – “Use bracket support in urban areas” is not a sentence and reads – like other parts of this new section – like an instruction manual. Please revise.

Line 293 – as a result of replacing the line graph with bar chart – much better by the way – you now leave text saying “contour curve” which now needs to be changed.

Line 472 – this big paragraph should be split – maybe when you talk about females’ responses.

Line 522 – be careful about how much confidence you show in Ulrich’s 1984 experiment – it is cited a lot but is not as clear-cut as you make it sound and more of a suggestion as to the implications.

Reviewer 4 Report

i think the current version is ready to be published 

Author Response

Comments and Suggestions for Authors

1.i think the current version is ready to be published 

Response: Many thanks for giving us an opportunity to revise our manuscript (Ms. No.: ijerph-774441). We appreciate for your acceptable and efforts for our manuscript. Although your comments are positive, we have also made some variations to enhance the quality of our manuscript, and we have polished our manuscript language. which were marked red in the modified version. We hope that the revisions and accompanying responses have made our manuscript suitable for publication in International Journal of Environmental Research and Public Health.

Once again, we thank you for the valuable advice.
